# Clinical Alarms in a Gynaecological Surgical Unit: A Retrospective Data Analysis

**DOI:** 10.3390/ijerph20054193

**Published:** 2023-02-26

**Authors:** Juho O. Jämsä, Kimmo H. Uutela, Anna-Maija Tapper, Lasse Lehtonen

**Affiliations:** 1Faculty of Medicine, University of Helsinki, 00014 Helsinki, Finland; 2GE Healthcare, 00510 Helsinki, Finland; 3Hyvinkää Hospital, Helsinki and Uusimaa University Hospital District, 05850 Hyvinkää, Finland; 4Diagnostic Center, Helsinki University Hospital, 00260 Helsinki, Finland

**Keywords:** alarm fatigue, anaesthesiology, clinical alarms, gynaecological surgery, monitor customisation, patient safety

## Abstract

Alarm fatigue refers to the desensitisation of medical staff to patient monitor clinical alarms, which may lead to slower response time or total ignorance of alarms and thereby affects patient safety. The reasons behind alarm fatigue are complex; the main contributing factors include the high number of alarms and the poor positive predictive value of alarms. The study was performed in the Surgery and Anaesthesia Unit of the Women’s Hospital, Helsinki, by collecting data from patient monitoring device clinical alarms and patient characteristics from surgical operations. We descriptively analysed the data and statistically analysed the differences in alarm types between weekdays and weekends, using chi-squared, for a total of eight monitors with 562 patients. The most common operational procedure was caesarean section, of which 149 were performed (15.7%). Statistically significant differences existed in alarm types and procedures between weekdays and weekends. The number of alarms produced was 11.7 per patient. In total, 4698 (71.5%) alarms were technical and 1873 (28.5%) were physiological. The most common physiological alarm type was low pulse oximetry, with a total of 437 (23.3%). Of all the alarms, the number of alarms either acknowledged or silenced was 1234 (18.8%). A notable phenomenon in the study unit was alarm fatigue. Greater customisation of patient monitors for different settings is needed to reduce the number of alarms that do not have clinical significance.

## 1. Introduction

Alarm fatigue is a phenomenon whereby medical staff become desensitised to patient monitoring alarms. For example, the Joint Commission recognises this phenomenon as a patient safety concern [1]. Alarm fatigue is also officially defined in the global standard for medical devices [2]. This fatigue may lead to a slower response time or total disregard of alarms and therefore affects patient safety. This phenomenon has been widely studied, although universal solutions are lacking. However, the need for customised solutions for units or individual patients is widely recognised [1].

The reasons behind alarm fatigue include the vast number of alarms that go off and their poor positive predictive value (PPV). Evidence for the exact correlation between alarm load and alarm fatigue and its magnitude is lacking; however, it is generally accepted that an increased alarm load likely increases alarm fatigue. Additionally, there is no consensus on how alarm fatigue should be measured [3]. Recently, a study in an emergency unit reported alarms sounding nearly continuously [4]. This phenomenon is complex and affected by various issues, including organisational procedures, staff workload, patient characteristics, and manufacturers’ alarm settings, which are purposely set to a low specificity value [3].

The majority of relevant research concentrates on intensive care unit (ICU) environments, operating room (OR) environments, and others [5,6,7]. One study reported 1.2 alarms per minute during cardiac surgery, 80% of which had no therapeutic consequences [8]. Another study found that during general surgical procedures, one alarm rang every 2.9 min, with an overall PPV of 36%. Alarm types, numbers, and their clinical significance vary in relation to the phase of anaesthesia [9].

We have previously shown that the customisation of monitoring devices in an emergency department could reduce the number of alarms that do not have any clinical impact [4]. This current study examined clinical alarms in the Surgery and Anaesthesia Unit of the Women’s University Hospital in Finland. The key objectives of this study were to evaluate the types and frequencies of patient monitoring device clinical alarms and to study the differences in alarm types between procedures. Clinical alarms constitute all alarms required to reduce risks to the patient. Clinical alarms include both physiological alarms related to patient status and technical alarms related to monitoring status. The secondary objectives were to indirectly evaluate alarm fatigue and compare these findings with those of our previous study in an emergency unit with identical monitors [4].

## 2. Materials and Methods

Ethical approval for this study (Ethical Committee N° HUS/486/2018) was provided by the Ethical Committee II of Helsinki University Hospital, Helsinki, Finland, on 6 June 2018. The research was conducted under a national process that obviated the need for written informed consent. The study was performed in the Women’s Hospital, Helsinki, during autumn 2019, by retrospectively collecting data from patient monitoring device clinical alarms and patient characteristics from surgical operations. All participants were female. The unit performs various types of elective and acute gynaecological and obstetric operations, except those relating to cancer. The average number of patients treated per month is 600; only emergency procedures are performed on weekends.

During the study, 10 operating rooms were in use. Of these, two were specifically allocated for emergency operations (A and B); other rooms were used if needed. In addition, two rooms (C and D) were specifically allocated for caesarean sections (C-section). Patient data included the operating room, date of operation, patient age, weight, height, BMI, list of surgical procedures, and surgical position. However, only a patient’s age and the list of surgical procedures were recorded for every operation by the operating surgical team. Usually, emergency operations had less information recorded, but the detailed reasons for the lack of data were unknown to us. The Finnish classification of surgical procedures is mostly based on the Nordic Classification of Surgical Procedures (NCSP).

Induction and, during general anaesthesia, emergence, are performed by an anaesthesiologist, who may adjust the alarm limits according to the clinical evaluation. During C-sections, an anaesthesiologist is present until the baby is delivered. In other cases, a nurse anaesthetist monitors the maintenance phase of the procedure and performs the emergence phase. The nurse is allowed to change monitor alarm settings, if deemed clinically necessary.

Patient monitors in the department were of type CARESCAPE B850. Alarm data were collected remotely by the monitor manufacturer (General Electric) over four consecutive weeks (9 September to 6 October 2019). The monitored modalities included electrocardiography (ECG), respiratory rate (RR) from an impedance sensor, invasive and noninvasive blood pressure (NIBP), and pulse oximetry (SpO_2_). Monitors had identical default alarm settings, except in rooms C and D, where alarms were set to a more sensitive preset value that enabled additional ECG alarms. Alarms produced by the monitors were categorised by built-in algorithms into three priority levels: high, medium, and low. Alarms sounded and remained active until they were either silenced or acknowledged with the respective button by the user, or if the sensed parameter spontaneously returned to a value outside the alarm threshold. A silenced alarm remained active so that it became inaudible but remained visible on the screen. An acknowledged alarm was deactivated. If multiple alarms co-occurred, the one with the highest priority remained audible until the situation changed. Informative monitor messages, which had no alarm priority and were not required as patient risk mitigation, were excluded from this study. These messages, for example, notified of user actions or were related to minor technical issues. Some of the messages, such as arrythmia paused, in some situations escalated as an alarm according to built-in algorithms.

An alarm is set off after an averaged parameter value is sensed to have crossed a preset threshold and a possible time delay has passed. In providing parameter values, the monitors had the following averaging settings: heart rate (HR) used the median of the last 12 s; RR from the impedance sensor used an average of four breaths; and SpO_2_ used the average of the last 12 s. The default alarm threshold settings were RR high; 25 breaths min^−1^; SpO2, 90%; bradycardia/HR low, 40 bpm; and tachycardia/HR high, 150 bpm. Alarm delays in the monitors were as follows: bradycardia, tachycardia, and HR low/high were given without time delay; SpO_2_ and RR alarms used a delay of 5 s. NIBP (systolic) is a one-time measurement, with no time delay, and a low threshold of 80 mmHg and a high threshold of 180 mmHg. The settings described here were identical to those used in a study in an emergency unit [4].

Regarding alarm duration, we chose 10 s as a cut-off value for momentary alarms; these alarms were transient in nature and likely spontaneously disengaged. It is reasonable to assume that no therapeutic action, such as drug administration, could be completed in under 10 s, thus resulting in the normalisation of vitals and termination of the given physiological alarm. On the other hand, many actions in response to technical alarms, such as reattaching an SpO_2_ sensor, might be completed in under 10 s with the consequent normalisation of the SpO_2_ sensor alarm. This 10 s value is the same as that used in our previous study [4]; it is a more conservative value than 16 s, which was used in an observational study on cardiac surgery [10]. A momentary alarm is not to be understood as a nonrelevant alarm. For example, many physiologic ECG-related alarms, such as a missing beat, may be transient in nature but still clinically relevant, if they are confirmed as true alarms in the ongoing clinical situation.

The procedure durations were not directly available in our data. We calculated the average time taken from the onset of the first alarm in a procedure to the time the monitor was reset by the clinician at the end of a procedure. Calculated durations of under five minutes were excluded.

From the collected data, we descriptively analysed patient characteristics, surgical procedures, and the number, types, values, and durations of monitor alarms. We also analysed the differences in alarm types between weekdays and weekends because only emergency procedures were performed on weekends. This analysis used a chi-squared test with a cut-off *p*-value of 0.05. However, only alarms that were produced by SpO_2_, NIBP, or ECG sensors were included in the statistical analysis, because these sensors were always attached to a patient, regardless of patient characteristics or type of procedure. Alarm fatigue was assessed indirectly from the alarm load and perceived alarm reliability, which is a common method in the literature [3].

## 3. Results

### 3.1. Patient Characteristics

During the study period, 615 patients were treated in the 10 operating rooms. Of these rooms, one room was used only twice, and the alarm data from one monitor were incomplete; these operating rooms were excluded. We analysed data from 8 monitors and 562 patients, of whom 101 patients had no record of the operating room in which they were treated. The average patient characteristics were as follows: age 43.6 (±16.9) (all) and BMI 27.6 (±6.9) (provided for 359 patients, of which 350 were treated on weekdays). A total of 509 patients, with an average age of 44.5 (±17.3), were treated on weekdays; 53 patients, with an average age of 34.8 (±8.7), were treated on weekends.

### 3.2. Clinical Alarms

The analysed data illustrate that the monitors (95% CI 6535 to 6607, see Methods) produced 6571 alarms, that is, 11.7 alarms per patient. Of all the alarms, 4698 (71.5%) were related to the technical status of the monitor, and 1873 (28.5%) were related to the physiological status of the patient. In the latter type of alarm, the threshold differed from the preset value in 35 (1.9%) alarms. Alarms were either acknowledged or silenced 1234 (18.8%) times. The number of pre-silenced, i.e., inaudible but active, alarms was 292 (4.4%).

A total of 5947 alarms were set off on weekdays and 624 on weekends. Thus, the number of alarms per patient on weekdays was 11.7 and 11.8 on weekends.

The most common physiological alarm types were SpO_2_ low, 437 (23.3%), and NIBP sys low, 350 (18.7%). The average duration of these alarms (inaudible and audible) was 27 s (±54 s) and 2 min 22 s (±2 min 4 s), respectively. The most common technical alarms were leads off, at 1462 (31.1%), and SpO_2_ probe off, at 1430 (30.4%). The average duration of those alarms was 33 s (±55 s) and 43 s (±1 min 19 s), respectively. Table 1 displays the relevant data.

The number of momentary alarms was 1863 (28.4%), and their proportion among the alarm types varied. Table 1 displays the data for the most common alarm types.

Of the momentary alarms, 871 (46.8%) were physiological alarms that likely disengaged without manual intervention with the monitor. A total of 992 (53.2%) were technical alarms. Of the technical alarms, 287 (28.9%) were types typically related to transient artifacts (e.g., arrhythmia paused), likely disengaging without manual intervention, and 705 (71.1%) were types typically requiring manual intervention to be disengaged (e.g., ECG leads off). Of the SpO_2_ low alarms, 131 (30.0%) had a value above 88%; of the NIBP sys low alarms, 239 (68.3%) had a value above 75 mmHg; and of the bradycardia alarms, 164 (82.4%) had a value above 35 bpm. Table 2 displays alarm values by alarm type for the most common alarm types.

The most common alarm types did not differ among the operating rooms. The number of alarms varied between 525 and 1710. The two operating rooms in which most of the C-sections were carried out were exceptions; certain alarms were produced only in these two operating rooms, with 382 set off in total (17.1% of alarms in rooms C + D, 5.8% of all unit alarms). All these alarms were ECG-related, and the missing beat alarm was among the most frequent, with 96 set off (5.6% of room C alarms) in room C and 90 (17.1% of room D alarms) in room D. Of these ECG alarms, 351 (91.9%) were momentary.

### 3.3. Clinical Procedures

Altogether, 2389 procedures were recorded for the operations. Of these, 1110 were anaesthetic (WX codes in the NCSP), 950 operational (i.e., type of surgery), and 329 categorised as other (e.g., imaging, urgency). On average, each patient had 4.3 procedures per operation. The most common anaesthetic procedures were general anaesthesia, at 321 (28.9%), and local anaesthesia, at 208 (18.7%). The most common operational procedures were C-section, at 149 (15.7%), and anterior colporrhaphy with sutures for primary cystocele, at 53 (5.6%).

C-sections were mainly performed in two specific operating rooms, with 55 (36.9%) performed in one and 41 (27.5%) in the other. A total of 53 (35.6%) C-sections had no record of the operating room in question.

There were statistically significant differences in alarm types between weekdays and weekends. Only alarms set off by SpO_2_, NIBP, and ECG sensors were included. SpO_2_ probe off (*p* = 0.001) and bradycardia (*p* < 0.001) alarms were more common on weekdays, and check NIBP alarms (*p* < 0.001) were more common on weekends. Table 3 shows the results. It was not possible to study the statistical differences in the proportion of acknowledged or silenced alarms between weekdays and weekends, as it was not possible to match those actions with alarms from specified sensors. Regarding all alarm types, the number of acknowledged or silenced alarms was 1114 (18.7%) on weekdays and 120 (19.5%) on weekends.

Differences were statistically significant regarding the execution of procedures on weekdays and on weekends. In general, more operational procedures were recorded on weekdays (*p* = 0.020). Of the anaesthetic procedures, local anaesthesia was more commonly administered on weekdays (*p* < 0.001) and lumbar epidural anaesthesia on weekends (*p* < 0.001). No difference existed in the proportion of general anaesthesia (*p* = 0.052). Of the operational procedures, C-section was more common on weekends (*p* < 0.001). Table 4 shows the results.

The calculated average duration of a procedure was 1 h 30 min 43 s (±57 min 25 s). On weekdays, the average duration was 1 h 34 min 22 s (±58 min 21 s), and on weekends, it was 56 min 3 s (±30 min 47 s). Thus, the number of alarms per operation per hour was 7.4 on weekdays and 12.6 on weekends.

## 4. Discussion

The main finding of our study was the substantial number of alarms set off in the operating room during a gynaecological procedure: 11.7 per patient. Further analysis revealed that 7.4 alarms were set off per operation per hour on weekdays and 12.6 on weekends, and as stated previously, only emergency procedures were performed on weekends. A limited number of studies focus on clinical alarms and alarm fatigue in the context of surgery; however, we will compare our findings with these studies. One study recorded 25 alarms per hour during a gynaecological procedure [9]. Another study recorded 7.5 audible alarms per patient during four different kinds of surgical procedures (including gynaecological procedures) [11].

In addition to a high number of alarms, a poor PPV is hypothesised to be a cause of alarm fatigue [1,3,5,6,7]. This study was not observational; therefore, we were unable to study the clinical correspondence of the alarms. However, there was some evidence of poor perceived alarm reliability. Firstly, in only 35 (1.9%) of the physiological alarms, the alarm threshold was modified from the preset value. This is considered a sign of alarm irrelevance in other studies [9]. Secondly, only a few alarms, 19%, were either silenced or acknowledged by the clinician. Possible relevant explanations for this include that the alarm was spontaneously disengaged, and thus, the need to take action was seen as questionable (28% of all alarms were momentary); the alarm was missed by the clinician, which is a direct consequence of alarm fatigue; the clinician knowingly chose not to acknowledge the alarm, as the clinician may have been busy with clinical intervention or may have interpreted the alarm as not relevant.

In other studies, the proportion of clinically irrelevant alarms in the context of surgery ranges from 64% to 80% [8,9,10]. One study found that 28% of alarms during anaesthesia led to a clinical intervention [11]. Of these mentioned studies, two [8,11] included technical and physiological alarms, and two [9,10] focused on physiological alarms. One study reported that only 18% of technical and 5% of physiological alarms led to an action by the clinician within 5 min [12]. In our study, the majority of alarms were related to the technical status of the monitoring device. This finding differs from previous findings [8,11]. Many studies focus on physiologic alarms, while it is possible that technical alarms more often require action from the attending nurse and that technical alarms are thus relevant to alarm fatigue [12]. It is hypothesised that even inaudible technical alarms are relevant to alarm fatigue [13]. Previous research has concluded that alarm types and frequencies differ between the phase of anaesthesia [9,11]. A higher alarm frequency was found during emergence than during the maintenance phase [9]. Patient movement and therapeutic manoeuvres increase the likelihood of an alarm flood [6]. It is likely that our findings are relevant to these issues. To conclude, our study suggests that alarm fatigue is a real and notable phenomenon occurring in gynaecological operating rooms as well.

Regarding physiological alarms, our study extends data from earlier studies; the most common alarms during surgery are related to low oxygen saturation, low blood pressure, and a low heart rate [8,9,11].

Most of the momentary alarms (62%) in our study disengaged without manual intervention with the monitor. Further analysis revealed that an estimated 29% of technical momentary alarms disengaged without manual intervention with the monitor. The classification between user-ended and self-corrected momentary technical alarms was based on accumulated data and knowledge about the typical reasons for alarms from the monitor manufacturer. Our data do not reveal which interventions the clinician performed to a patient in any given situation, and if these therapeutic or measurement corrective actions disabled the alarm. As noted earlier, momentary alarms are not to be understood as irrelevant. This issue is complex, and the whole clinical situation should be taken into account, especially in regard to transient-like ECG-related physiological alarms with patients who possess a known cardiac condition. In two of the operating rooms, ECG alarms were set to an optional setting designed for cardiac patients. The use of this setting in our study unit is clinically questionable as it was used in only some of the monitors, regardless of the actual cardiac condition of a patient, and as the majority of the alarms were momentary.

As noted earlier, according to previous studies, most clinical alarms are irrelevant or non-actionable. This applies for both physiological and technical alarms [8,9,10,11,12]. It is reasonable to assume that the same issue applies to momentary alarms as well, but further studies are needed. We conclude that eliminating momentary alarms by incorporating time delays should be considered, in order to reduce the number of non-actionable alarms. In particular, physiologic momentary alarms of noncardiac patients should be critically evaluated.

Using additional time delays and changing alarm thresholds are common methods to reduce the number of alarms and alarm fatigue [3]. According to our data, if the alarm threshold limits of SpO2 low at 90%, NIBP systolic low at 80 mmHg, and bradycardia at 40 bpm were set to 88%, 75 mmHg, and 35 bpm, respectively, the number of alarms might decrease by 30%, 68%, and 8%, respectively. If an additional 10 s time delay was introduced, the number of these alarms might be reduced by 42%, 11%, and 55%, respectively. Further, if these changes were applied simultaneously, the estimated reduction in the number of alarms would be 55%, 72%, and 97%, respectively. However, it should be noted that these values are hypothetical maximums. It is not possible to conclude from our data which alarms would have progressed further beyond the threshold limit, and which would have returned to a normal value. Thus, it cannot be said that, e.g., changing the SpO2 threshold to 89% would eliminate all alarms produced at the 90% value.

In addition, the clinical safety of these threshold changes is debatable. A survey study among anaesthesiologists concluded that reasonable clinical values for the previously mentioned alarms averaged at 88%, 86 mmHg, and 48 bpm, respectively [14]. Another survey study in a cardiac postoperative clinic found that anaesthesiologists’ alarm limits for therapeutic intervention were NIBP systolic low at 70 mmHg and bradycardia at 46 bpm; the duration of alarms before intervention was 23 s and 30 s, respectively [15]. It can be concluded that a 10 s time delay, rather than threshold changes, would likely be a more effective method of reducing the number of alarms.

We found that alarms differ between types of procedures as well as between units. An earlier study in an emergency clinic with identical monitors and settings recorded 6.3 alarms per hour per patient, of which 53% were momentary, and the most common types were RR high, SpO_2_ low, and SpO_2_ probe off [4]. Arguably, the use of the same default settings in two very different kinds of environments is clinically unreasonable.

Our study also discovered statistically significant differences in alarms between emergency and other types of procedures. In addition, the number of alarms per operation per hour was higher during emergency operations (12.6 vs. 7.4). These differences may be associated with the type of operation, type of anaesthesia, or patient age. A higher alarm load during emergency operations suggests increased alarm fatigue and decreased patient safety. This issue should be further studied.

We have previously shown that, compared to nonserious incidents, serious incidents are more common in operational units and are related to monitoring, medical equipment, and equipment operation [16]. The current study unit was among the units considered in our previous study [16]. To summarise, our study confirms earlier suggestions that alarms differ between units and patient populations, and studying individualised alarm settings is advisable to enhance patient safety [1].

This study had some limitations. Most notably, this was not an observational study, and thus, the clinical significance of alarms and clinicians’ responses were not assessed. These are important factors in the development of alarm fatigue. Secondly, this was a single-clinic, single-monitor-type study; consequently, the application of the results to other settings is limited. Additionally, our results are suggestive, as we studied the differences between operations on weekdays and on weekends, instead of between types of operations. Had we been able to consider the latter, the results would likely have been even more significant. In addition, due to separate data sets, we were unable to match individual patient characteristics with corresponding clinical alarms.

## 5. Conclusions

We found that the number of patient monitoring alarms in a gynaecological surgical unit was 11.7 per patient. Implementing additional time delays, rather than changing the alarm thresholds, may be a more efficient and reasonable method of reducing the number of alarms. The type, number, and duration of alarms differed between our study unit and an emergency room with identical patient monitor settings. Alarms also differed for the types of procedures in the study unit. Emergency operations were particularly correlated with a higher alarm load. Alarm fatigue seemed to be a notable phenomenon in the study unit, and greater customisation of patient monitors for different settings is thus needed to reduce the number of alarms without clinical significance.

## Figures and Tables

**Table 1 ijerph-20-04193-t001:** The 10 most common alarm types out of 53.

Alarm Type	Alarm Threshold Value	Number of Alarms	Percentage of Alarms	Average Duration (min:s ±SD)	Number of Momentary Alarms	Percentage of Momentary Alarms
ECG leads off (T)	N/A	1462	22.2%	0:33 ± 0:50	218	14.9%
SpO_2_ probe off (T)	N/A	1430	21.8%	0:43 ± 1:19	254	17.8%
SpO_2_ low, (%) (P)	90	437	6.7%	0:27 ± 0:54	183	41.9%
NIBP systolic low, (mmHg) (P)	80	350	5.3%	2:22 ± 2:04	40	11.4%
Check NIBP sensor (T)	N/A	294	4.5%	0:40 ± 0:34	52	17.7%
No entropy sensor (T)	N/A	254	3.9%	4:57 ± 6:02	26	10.2%
Arrhythmia paused (T)	N/A	227	3.5%	0:28 ± 0:27	66	29.1%
NIBP auto-stopped (T)	N/A	210	3.2%	1:07 ± 1:37	50	23.8%
Bradycardia, (bpm) (P)	40	199	3.0%	0:31 ± 0:57	109	54.8%
Check pulse oximetry probe (T)	N/A	196	3.0%	0:15 ± 0:12	77	39.3%
Total (TOP 10)		5059	77.0%	0:57 ± 1:18	1075	21.2%
Total (all alarms)		6571	100%	0:59 ± 2:23	1863	28.4%

Momentary alarm, duration less than 10 s; (P), physiological alarm; (T), technical alarm; NIBP, noninvasive blood pressure; N/A, not applicable.

**Table 2 ijerph-20-04193-t002:** Alarm values (initial) and the corresponding number of alarms.

Alarm Values by Alarm Type	Number of Alarms	Number of Momentary Alarms	Percentage of Momentary Alarms
SpO_2_ low (%)	437	183	41.9%
86	39	13	33.3%
87	45	19	42.2%
88	46	20	43.5%
89	88	46	52.3%
90	43	29	67.4%
NIBP sys low (mmHg)	350	40	11.4%
76	27	6	22.2%
77	37	5	13.5%
78	49	5	10.2%
79	53	4	7.5%
80	73	6	8.2%
Bradycardia (bpm)	199	109	54.8%
36	16	12	75.0%
37	11	8	72.7%
38	19	12	63.2%
39	21	9	42.9%
40	97	38	39.2%

NIBP, noninvasive blood pressure. The default alarm thresholds were SpO_2_ low, 90; NIBP low, 80; and bradycardia, 40.

**Table 3 ijerph-20-04193-t003:** Differences in alarm types between weekdays and weekends for the three most common physiological and technical alarms using chi-squared test. (Only SpO_2_, ECG, and NIBP sensor alarms were included).

Alarm Type	Number of Alarms, Weekdays	Percentage of Alarms, Weekdays	Number of Alarms, Weekends	Percentage of Alarms, Weekends	Χ^2^	*p*
ECG leads off (T)	1327	25.7%	135	23.2%	1.7833	0.182
SpO_2_ probe off (T)	1317	**25.5%**	113	19.4%	10.5007	0.001
SpO_2_ low, (%) (P)	384	7.4%	53	9.1%	2.0418	0.153
NIBP systolic low, (mmHg) (P)	316	6.1%	34	5.8%	0.0756	0.783
Check NIBP sensor (T)	233	4.5%	61	**10.5%**	38.2217	<0.001
Bradycardia, bpm (P)	198	**3.8%**	1	0.2%	19.9424 *	<0.001 *
Total (all alarms)	5164		583			

(P), physiological alarm; (T), technical alarm; NIBP, noninvasive blood pressure; *, Yates’s correction applied; for ease of reading, the higher percentage is bolded regarding alarms for which a statistical difference was found.

**Table 4 ijerph-20-04193-t004:** Differences in reported procedures between weekdays and weekends, using chi-squared test.

Procedure	Number of Procedures, Weekdays	Percentage of Procedures, Weekdays	Number of Procedures, Weekends	Percentage of Procedures, Weekends	Χ^2^	*p*
General anaesthesia (A)	302	29.7%	19	20.2%	3.7868	0.052
Local anaesthesia (A)	206	**20.3%**	2	2.1%	17.4365 *	<0.001 *
Lumbar epidural anaesthesia (A)	143	14.1%	31	**33.0%**	23.2607	<0.001
Repeated or continuing anaesthesia shots through a catheter (A)	133	13.1%	22	**23.4%**	7.618	0.006
Spinal anaesthesia (A)	125	12.3%	11	11.7%	0.0289	0.865
C-section (O)	124	13.9%	25	**42.4%**	33.8829	<0.001
Anterior colporrhaphy with sutures for primary cystocele (O) **	53	5.9%	0	N/A	N/A	N/A
Repair of partial rupture of perineum (O) ***	6	0.7%	5	**8.5%**	23.0034 *	<0.001 *
Anaesthetic procedures (all)	1016	46.1%	94	50.5%	1.3463	0.246
Operational procedures (all)	891	**40.4%**	59	31.7%	5.4504	0.020
Other procedures (all)	296	13.4%	33	17.7%	2.6778	0.102
Total	2203		186			

The percentages are calculated from the subtype totals, e.g., general anaesthesia % of anaesthetic procedures; (A) anaesthetic procedure; (O) operational procedure; *, Yates’s correction applied; ** this was the second most common operational procedure on weekdays; *** this was the second most common operational procedure on weekends; N/A, not applicable; for ease of reading, the higher percentage is bolded for procedures for which a statistical difference was found.

## Data Availability

The patient data presented in this study are available on request from the corresponding author. The data are not publicly available due to privacy restrictions. Restrictions apply to the availability of monitor data. Data were obtained from General Electric and are available from the corresponding author with the permission of General Electric.

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
