# Peer review of "Clinical Alarms in a Gynaecological Surgical Unit: A Retrospective Data Analysis"

_ijerph, 2023, doi:10.3390/ijerph20054193_

Round 1
Reviewer 1 Report
L10: There is not yet a commonly accepted definition of “alarm fatigue”, maybe rephrase “…can be understood as…”
L29: see comment to l10
L36 “increased alarm load increases alarm 36 fatigue” it may be assumed, that this is the case, but in the absence of both a clear definition of “alarm fatigue” and, consequently, a metric for measuring it, this statement is too bold.
L38: “reported a near constant alarm sound load”: Constant can also be constantly low or even constantly zero. Rephrase.
L88: “Alarms sounded and staid active until they were either silenced or acknowledged […]. A silenced alarm remained active[…]” Does a silenced alarm stay active or not?
L104: I fail to understand the concept of “momentary alarm”. I assume, that it is meant to describe transient, self-correcting situations. If this is correct, then one would need to ensure that the definition (alarm duration < 10s) indeed excludes (almost) all manually terminated alarms. More importantly, the significance of the concept remains foggy. Is the reader to assume that “momentary alarms” are non-actionable? At least for clinical alarms one might argue that even a short episode of threshold violation might be of importance. Another problem here is that some clinical alarms are not threshold alarms, but rather “events” (e.g. many arrhythmia alarms, but also apnea or asystole). Are these per definition momentary? L157 mentions “missing beat”, which is of this “event type”. Do they auto-terminate? I would ask the authors to clearly explain what the meaning and the importance of “momentary alarms” are, and how they made sure that the time threshold of 10s is a good enough criterion to determine whether an alarm is “momentary”.
L140: “The average duration of those alarms were…” As it is not clear whether silencing keeps an alarm active or not (see comment to L88), it is also not clear how the alarm duration is calculated: Is it the duration of the audible signal, or of the visual indicator? This makes it hard to attribute value to the given numbers for duration. I would suggest to differentiate between self-terminating and humanly-terminated alarms. For the latter, the alarm duration can be interpreted as “reaction time”, which is an important metric. As given, the alarm duration is just one parameter to estimate the cognitive load the operator is experiencing, and even this is very vague, if one doesn’t know whether this load is induced by the auditory or the visual signal, or both.
L147: Inconsistent naming: The table 1 has “pulse oximetry low”, but the text refers to the same alarm by “SpO2 low”.
L171: “Differences in alarm types were statistically significant between weekdays and 171 weekends.” To draw meaningful conclusions from this fact, one would need to know whether the same equipment was used in the procedures on weekends and during the week. It seems plausible to assume that during emergency operations, all equipment was used, while for minor and elective procedures, one might, for instance, not have used invasive blood pressure measurements. Only if the two subpopulations have – by and large – the same utilization of available measurement equipment, the comparison makes sense.
L208 “This study was not observational; therefore, we were unable to 208 study the clinical correspondence of the alarms.”. While this is true, one should also not that more that two thirds of the alarms were technical anyway. Taking only the data listed in Table 1, one would calculate that 65% of the listed alarms were a) technical and b) non-momentary (81% of the technical alarms were non-momentary), i.e. most likely required a human intervention (e.g. reattaching sensor or cable). With these numbers, the PPV of a technical alarm would be 0.81 and the PPV of all alarms at least 0.65. This argument is not to negate the possibility that many of the observed alarms were non-actionable, but rather to support the claim that technical alarms and clinical alarms should be analyzed differently. Taking the term “PPV” in the strictest sense, the PPV of a technical alarm is always 1, as there is a condition that requires intervention (even though it may be transient and self-correcting).
As the key objective of the study was stated as “to evaluate the types and frequencies of patient monitoring device clinical alarms”, I would suggest to emphasize the difference between clinical and technical alarms much stronger throughout the paper, maybe even discussing them separately. Definitions vary between the two categories, as explained above, the associated cognitive load is different and also their behavior is different (e.g. repeats, possibility to deactivate and the like).
L211: “These (technical) alarms may not require, or may be interpreted as not requiring, clinical intervention, which can lead to alarm fatigue” I would seriously doubt this. Clinical alarms which are non-actionable most likely lead to alarm fatigue, but technical alarms, which are – as discussed above – almost per definition actionable, even though they do not require a clinical intervention, can not be regarded as having the same effect. The publication cited by the authors to support the above claim (Hravnak M, Pellathy T, Chen L, et al. A call to alarms: current state and future directions in the battle against alarm fatigue. J 318 Electrocardiol 2018;51:44-8) very clearly focusses on clinical alarms, and can not be interpreted as saying that technical alarms would contribute to alarm fatigue.
L236: “According to our data, if the alarm threshold limits of SpO2 low 90%, NIBP systolic low 80 mmHg, and Bradycardia 40 bpm, were set to 88%, 75 mmHg and 35 bpm respectively, the number of alarms could decrease 238 by 30%, 68% and 8% respectively.” I don’t agree with this conclusion, at least it can not be drawn from the presented data. Take for example the case of SpO2 alarms. In 30% of the cases, the trigger value vas above 88%. But this does not necessarily mean that during the alarm the value did not go to deeper levels. But even if this were the case, and then the authors should state this explicitly, one cannot assume that setting the limit to 88% would have avoided the alarm. In the “normal case” the alarm alerted the anesthetist to the condition, and he or she reacted accordingly, bringing the value back to above 92%. In many cases, the SpO2 value would have decreased further without the intervention, alarming at 88% instead of at 92%. One can only speculate on how often the SpO2 would have self-corrected without an intervention, and only this proportion of alarms would have been avoided by adjusting the SpO2 low limit.
L261: “The use of this setting in our study unit is clinically questionable as it was used in only some of the monitors, and the majority of the alarms that 262 were thus produced were momentary.” As mentioned above, I don’t see a good argument why “momentary” alarms are not useful. In particular, in the mentioned setting of cardiac patients, where arrhythmia alarms are frequent, and most of these have the character of an event. At least the authors should explain whether or not “event alarms” can be momentary. After all, a “momentary” asystole of 9s duration shouldn’t be regarded as insignificant.
(Overall, I would suggest to have a native speaker of English proof-read the paper)
Reviewer 2 Report
The authors uderline the importance of clinical teamwork in any situation. The fact that alarm signs trasfer alert messages to clinicians may surely help to intercept a potentialo danger in patient management. These requisites are well detailed and confirm the importance to adopt a proactive approach to patient safety. In the specific field of maternal management the necessity of keeping alarms on and also exchange verbal comments are essentials of application of non technical skills in healthcare settings.
Author Response
Dear Reviewer,
we appreciate your encouraging and professional comments. We also find the discussed issues important and deserving of more attention.
Reviewer 3 Report
Dear Authors,
you have undoubtedly raised a very interesting topic, which is important both for the comfort of doctors' work and improving patient safety.
The entire manuscript is presented in a very clear and legible way. Materials and methods are described exhaustively and the database collected is extensive. The results are presented in detail and are understandable. The discussion is interestingly written and contains important findings and suggestions. Conclusions are concise, scientifically valid and supported by the results obtained.
On a minor note, I wonder why only the age of the patients and the list of surgical procedures were provided for every operation? What was the reason for the lack of other data? (Lines 70-71).
In lines 100-102 about the alarm threshold for NIBP measurements, it is not specified whether the pressure values refer to systolic, diastolic pressure or MAP. From the given values and the whole manuscript, it is easy to deduce that the authors were referring to systolic blood pressure, but shouldn't it be clarified here as well?
In conclusion, I would like to thank you for the opportunity to review such a good manuscript and congratulate the authors.
Author Response
Dear Reviewer,
we appreciate your encouraging comments.
In regard to note about L70. The reason for the lack of data was unknown to us. It had something to do with how the operating team made recordings in different procedures. This issue is now stated in the text (L77)
Regarding comment about NIBP (L100), we added a specifying notion that we refer to systolic blood pressure in our article (L113)
Round 2
Reviewer 1 Report
I very much thank the authors for taking my comments so seriously. While this is admittedly a biased view, the paper improved significantly.